# ViCO: A Training Strategy towards Semantic Aware Dynamic High-Resolution

## Abstract

Existing Multimodal Large Language Models (MLLMs) suffer from increased inference costs due to the additional vision tokens introduced by image inputs. In this work, we propose Visual Consistency Learning (ViCO), a novel training algorithm that enables the model to represent images of varying semantic complexities using different numbers of vision tokens. The key idea behind our method is to employ multiple MLP connectors, each with a different image compression ratio, to downsample the vision tokens based on the semantic complexity of the image. During training, we minimize the KL divergence between the responses conditioned on different MLP connectors. At inference time, we introduce an image router, termed Visual Resolution Router (ViR), that automatically selects the appropriate compression rate for each image patch. Compared with existing dynamic high-resolution strategies, which adjust the number of visual tokens based on image resolutions, our method dynamically adapts the number of visual tokens according to semantic complexity. Experimental results demonstrate that our method can reduce the number of vision tokens by up to 50% while maintaining the model's perception, reasoning, and OCR capabilities. We hope this work will contribute to the development of more efficient MLLMs. The code and models will be released to facilitate future research.

## 1 Introduction

Recent advancements in Multimodal Large Language Models (MLLMs) (Zhu et al., 2025; Chen et al., 2024c; Xiaomi, 2025; Kwai Keye et al., 2025; Anthropic, 2025; DeepMind, 2025; Wang et al., 2024b) have demonstrated remarkable performance across a wide range of tasks, showing tremendous potential for real-world applications. Despite these advancements, MLLMs still suffer from the increased inference costs due to the additional vision tokens introduced by image inputs. Taking InternVL3.5 (Wang et al., 2025b) as an example, under its default configuration (Ye et al., 2023), each image is divided into up to 13 patches (including one thumbnail). Each patch is then represented by 256 visual tokens, resulting in a maximum of 3,328 tokens per image. In real-world scenarios such as document understanding or video comprehension, models are often required to process multiple images simultaneously. In such cases, the visual component will serve as the main body of the token sequence and constitutes the primary source of inference cost.

To address these challenges, we propose Visual Consistency Learning (ViCO), which introduces semantic-level adaptivity in the number of visual tokens. As shown in Figure 1(a), given an image feature map, we introduce a Visual Resolution Router (ViR) that routes each patch to different compression rates: either a high-resolution representation with 256 tokens or a low-resolution representation with 64 tokens. These tokens are then concatenated with text tokens. The training procedure of ViCO consists of two stages: (1) *Consistency Training*: The model is trained to minimize the KL divergence between the responses conditioned on visual tokens with different compression rates. This encourages the model to generate accurate responses even when using highly compressed visual tokens, thereby improving its performance and robustness under compressed visual tokens. (2) *Router Training*: Although consistency training improves performance under high compression, inevitable information loss still causes a drop in performance. To mitigate this, we introduce ViR automatically selects the appropriate compression rate for each image patch. Those containing complex semantic information are represented using more tokens, while simpler patches are represented using fewer tokens. Compared with existing dynamic high-resolution strategies (Chen et al., 2024d; Wang et al.,

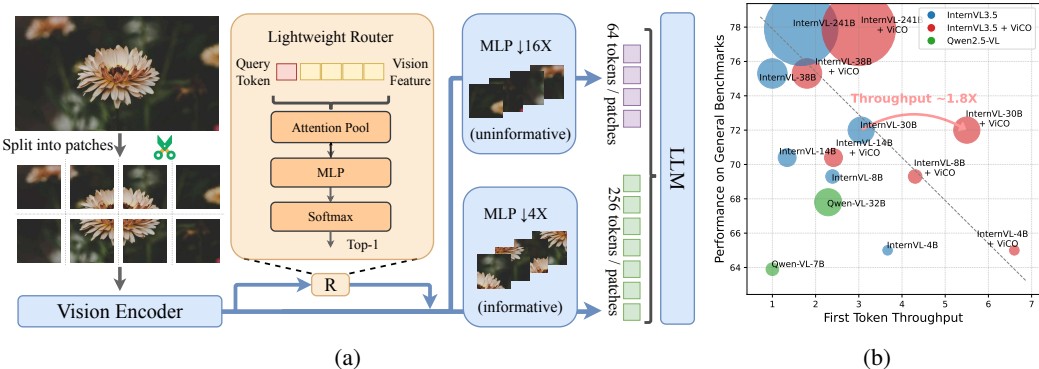

Figure 1: **Overall view of the ViCO pipeline and model performance.** (a) The ViCO inference pipeline, illustrating the image processing flow. (b) Bubble chart showing model performance on general benchmarks versus first token throughput of LLM. Bubble size is positively correlated with the number of model parameters. First token throughput is reported relative to InternVL3.5-38B, which is set as the baseline value of 1. Detailed experimental settings are provided in Section 4.6.

2024a), which adjust the number of visual tokens based on image resolutions, our method further determines the number of visual tokens allocated for each image patch at the semantic level. This fine-grained control enables substantial efficiency gains with minimal performance loss.

To evaluate the effectiveness of our method, we conducted extensive experiments on benchmarks spanning OCR, document understanding, video understanding, and multi-image reasoning. Experimental results show that our method can reduce the number of visual tokens by up to 50% while preserving the model's perception, reasoning, and OCR capabilities. As shown in Figure 1(b), our method maintains the original performance while improving the first token throughput of InternVL3.5 series (Wang et al., 2025b) across different model scales.

Our main contributions are as follows:

(1) We introduce a novel training strategy, termed Visual Consistency Learning, which minimizes the response distribution gap of the model between different visual token compression rates. This enables the model to effectively utilize highly compressed visual representations without significant performance loss.

(2) Building on ViCO, we develop a Visual Resolution Router that dynamically allocates visual tokens to image patches based on their semantic complexity. This provides fine-grained control over image patch representation and achieves a better trade-off between efficiency and accuracy.

(3) We perform large-scale experiments on benchmarks covering diverse multimodal recognition and reasoning tasks. Our results demonstrate that our method can halve the number of visual tokens while maintaining strong performance, which almost doubles inference throughput.

## 2 RELATED WORK

### 2.1 MULTIMODAL LARGE LANGUAGE MODELS.

With the development of large language models (Yang et al., 2025; OpenAI, 2025; Ouyang et al., 2022), multimodal large language models (MLLMs) have also made remarkable progress. To leverage LLMs and vision foundation models that have been pre-trained on unimodal datasets, a series of studies (Wang et al., 2024c; Liu et al., 2023a;d; Wang et al., 2025a; Li et al., 2023a) employ a connector to align the representational spaces of vision and language. This approach allows visual feature maps to be flattened and fed into LLMs as soft prompts, achieving strong performance through relatively low-cost incremental training. In addition, some works (Dubey et al., 2024; Luo et al., 2025) extend pre-trained LLMs by incorporating additional vision-language fusion layers. These layers enable the model to process tokens from different modalities with separate parameters, reducing the gradient conflicts across different modalities. However, the introduction of a large num-

ber of untrained parameters also brings additional training costs. More recently, several studies (Luo et al., 2024) have explored architectures without dedicated visual encoders. These models employ a unified Transformer to jointly process visual and textual information, eliminating the need for a separate vision encoder and fusion layer. Although MLLMs vary in their architectural designs, most of them adopt a dynamic high-resolution strategy (Chen et al., 2024c;d), which segments images into patches based on their resolution to enhance the model's perceptual capabilities. However, this approach requires a large number of tokens to represent each image, thereby increasing the inference cost of MLLMs. In this paper, we propose a dynamic resolution strategy that is compatible with this paradigm. Our method introduces semantic-level adaptivity in determining the number of visual tokens needed to represent each image patch. This reduces the number of visual tokens and consequently the inference cost, while maintaining nearly the same performance.

## 2.2 EFFICIENT VISION LANGUAGE MODELS.

Improving the efficiency of large vision-language models (LVLMs) has drawn increasing attention, with visual token compression emerging as one of the most widely explored directions. Early approaches, such as LLaMA-VID (Li et al., 2024) and DeCo (Yao et al., 2024), aim to reduce redundancy in visual inputs through context tokens or adaptive pooling mechanisms, thereby lowering computational cost while retaining essential information. Similarly, MADTP (Cao et al., 2024) further enhances efficiency by identifying redundant tokens across different modalities and selectively removing them based on feature alignment, enabling more focused processing of relevant visual features. A number of lightweight, training-free methods have been proposed to reduce token redundancy without additional training. For instance, FastV (Chen et al., 2024a) prunes tokens in the LLM based on attention scores, removing low-attention tokens, while similar approaches such as SparseVLM (Zhang et al., 2024) and ToMe (Bolya et al., 2022) also reduce redundancy through token merging or selection without requiring retraining. While these strategies provide tangible computational benefits, their effectiveness can be limited on vision-sensitive tasks such as OCR, where fine-grained spatial and textual details are crucial. In such scenarios, overly aggressive token compression may lead to loss of essential visual information, highlighting the need for methods that carefully balance efficiency with the preservation of detailed visual representations.

## 3 VISUAL CONSISTENCY LEARNING

To enhance recognition and perception capabilities, most existing MLLMs adopt a dynamic high-resolution strategy, which introduces a large number of visual tokens and greatly increases the inference cost of these models. In this work, we propose Visual Consistency Learning (ViCO), a novel training algorithm that enables the model to represent images of varying semantic complexities using different numbers of vision tokens.

## 3.1 CONSISTENCY TRAINING

As shown in Figure 2, in the Consistency Training stage, the model is trained to generate consistent output conditioned on different patch compression rates. In practice, we introduce an extra reference model, which is frozen during training. The trained model is required to minimize the KL divergence between its response distribution and that of the reference model. The training objective is formulated as follows:

$$\mathcal{L}_{\text{ViCO}} = \mathbb{E}_{\xi \sim \mathcal{R}} \left[ \frac{1}{N} \sum_{i=1}^{N} \text{KL}\Big( \pi_{\theta_{ref}} \left( y_i \mid y_{<i}, I \right) \, \Big\| \, \pi_{\theta_{policy}} \left( y_i \mid y_{<i}, I_\xi \right) \Big) \right], \tag{1}$$

where KL denotes the KL divergence and $\xi$ denotes the compression ratio of the patches in each image, which is uniformly sampled from $\mathcal{R} = [0, 1]$. The corresponding ratio of patches in image $I_\xi$ is represented as 64 tokens, while the others are represented as 256 tokens. The reference model always performs inference with visual tokens without any compression.

## 3.2 ROUTER TRAINING

As illustrated in Figure 2, the Router Training stage focuses on training the Visual Resolution Router (ViR), which is responsible for selecting an appropriate resolution for each input patch to balance

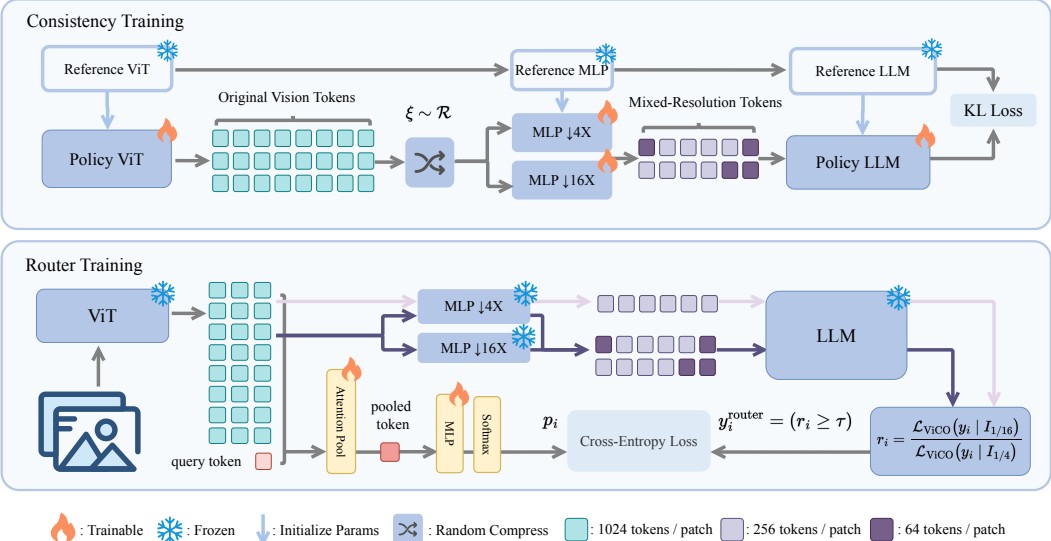

Figure 2: **Training procedure of Visual Consistency Learning (ViCO).** During the Consistency Training stage, the model aligns outputs under different compression rates. During the Router Training, the Visual Resolution Router (ViR) is trained to determine the appropriate compression for each patch based on its effect on model predictions.

efficiency and fidelity. ViR is implemented as a binary classifier and trained with a standard cross-entropy loss, while the main MLLM backbone remains frozen throughout this stage. To generate supervision signals for ViR, we first measure the effect of patch compression on the model's predictions. Concretely, for each patch, we calculate a loss ratio that quantifies how much the model's output degrades under compression. This ratio then serves as a guide for the router, indicating which patches can be safely compressed without significantly affecting overall performance. Specifically, for each patch, we compute a *loss ratio* defined as

$$r_i = \frac{\mathcal{L}_{\text{ViCO}}\big(y_i \mid I_{\frac{1}{16}}\big)}{\mathcal{L}_{\text{ViCO}}\big(y_i \mid I_{\frac{1}{4}}\big)}, \tag{2}$$

where $\mathcal{L}_{\text{ViCO}}$ denotes the consistency loss introduced in Section 3.1. This ratio captures the relative increase in loss caused by compressing the visual tokens, providing a principled measure of each patch's sensitivity to compression. Patches with low loss ratios can be safely compressed with minimal impact on the model's output, whereas patches with high loss ratios require higher resolution to preserve critical visual information. The binary ground-truth label for the router is then defined based on $r_i$:

$$y_i^{\text{router}} = \begin{cases} 0, & r_i < \tau \quad \text{(compression has negligible impact)} \\ 1, & r_i \geq \tau \quad \text{(compression has significant impact)}, \end{cases} \tag{3}$$

where $y_i^{\text{router}} = 0$ indicates that the patch can be compressed with a more aggressive strategy, and $y_i^{\text{router}} = 1$ indicates that the patch can be compressed with a more conservative strategy. To maintain a balanced training signal, we store historical $r_i$ values in a sliding window and dynamically set the threshold $\tau$ as the $k$-*th* percentile of these values. This approach typically results in roughly half of the patches being assigned to compression, which ensures a balanced distribution of target labels across patches. Notably, for each training sample, we randomly select a patch to compress and estimate its pseudo-label according to Equation 3. The loss is only computed on this patch.

In practice, the predicted router value for each patch is obtained by first extracting visual token features from the ViT backbone, aggregating them using attention pooling, and passing the resulting patch-level feature through a lightweight MLP. This process can be defined as:

$$p_i = \text{Softmax}\Big(\text{MLP}\big(\text{AttnPool}\big(\text{ViT}(I_i)\big)\big)\Big), \tag{4}$$

where $p_i^0$ and $p_i^1$ denote the predicted probabilities for the patch being assigned a high or low compression rate, respectively. The ViR is trained to match the ground-truth labels $y_i^{\text{router}}$ using standard cross-entropy loss.

## 4 EXPERIMENTS

We evaluate ViCO on InternVL3.5 models of various sizes, measuring both patch compression and performance retention relative to the original model. To validate the effectiveness of our approach across a wide range of domains, experiments are conducted on a diverse set of benchmarks, including general multimodal tasks (Section. 4.2.1), OCR-related benchmarks (Section. 4.2.2), and multi-image benchmarks and video benchmarks (Section. 4.2.3). OCR-related benchmarks are particularly sensitive to visual tokens, requiring fine-grained understanding of local details. Therefore, in comparative (Section. 4.3) and ablation (Section. 4.4) studies, we focus primarily on these benchmarks to provide a more discriminative evaluation.

### 4.1 EXPERIMENTAL SETTINGS

**Benchmarks.** We evaluate our approach on a diverse set of multimodal benchmarks. For OCR-related tasks, we use OCRBench (Liu et al., 2023c), ChartQA (Masry et al., 2022), and TextVQA (Singh et al., 2019), with InfoVQA (Mathew et al., 2022), DocVQA (Mathew et al., 2021), and AI2D (Kembhavi et al., 2016) additionally included in comparative and ablation studies. General multimodal tasks are evaluated on MMStar (Chen et al., 2024b), POPE (Li et al., 2023b), MME (Fu et al., 2023), MMBench V1.1 (Liu et al., 2023b), and RealWorldQA (Corp., 2024), while reasoning is assessed using the MMMU (Yue et al., 2023) benchmark. Multi-image understanding is measured on BLINK (Fu et al., 2024b), MMT-Bench (Ying et al., 2024), and MMIU (Ying et al., 2024), and video comprehension is tested on Video-MME (Fu et al., 2024a) under two settings: with subtitles and without subtitles.

**Training Details.** Our experiments are conducted in two stages: consistency learning and router learning. In the consistency learning stage, we adopt a maximum text length of 32K and use a global batch size of 256. The learning rate follows a cosine decay schedule from $4 \times 10^{-6}$ to $1 \times 10^{-7}$, and we use the AdamW optimizer. Additionally, we perform a warm-up for the newly initialized $16\times$ downsampling MLP. In the router learning stage, only the lightweight router is unfrozen for training, with the global batch size adjusted to 8. The routing threshold $\tau$ is set to the 60$^{\text{th}}$ percentile of the predicted scores. All experiments are conducted on H200 GPUs.

### 4.2 MAIN RESULTS

#### 4.2.1 RESULTS ON GENERAL BENCHMARKS

Table 1 presents the overall performance of InternVL3.5 models of different sizes (4B, 8B, 14B, 30B-MoE, 38B, and 241B-MoE) on a wide range of benchmarks, including OCRBench, ChartQA, TextVQA, MMStar, POPE, MME, MMBench V1.1, RealWorldQA, and MMMU. We compare the original models with their ViCO counterparts.

Across these benchmarks, each ViCO model retains over 99.6% of the original performance on average. Across all model sizes, the average retention is about 99.7%. For instance, the 8B model with ViCO achieves nearly identical scores to its baseline on POPE (88.7 vs. 88.4) with about 81% patch compression, together with strong results on MMStar, MMBench V1.1, and RealWorldQA, yielding an overall average of 99.6%. Similarly, the 38B and even the largest 241B-MoE models preserve strong performance across representative benchmarks, both maintaining around 99.6% relative to their original versions.Notably, for the MMMU, we adopt the thinking mode, which supports outputs up to 64K tokens. Even with such long outputs, the performance remains almost unchanged when using ViCO, further validating the method's stability and robustness. Overall, this high level of performance retention holds consistently as the model scales from 4B to 241B-MoE, demonstrating that ViCO is both effective and scalable.

Table 1: **Performance of ViCO on InternVL3.5 models across general benchmarks, including strongly OCR-related benchmarks.** When calculating the overall score, MME is normalized from 0–2800 to 0–100. RWQA refers to RealWorldQA.

| Model | OCRBench | ChartQA | TextVQA | MMStar | POPE | MME | MMBench V1.1 | RWQA | MMMU | Avg. |
|---|---|---|---|---|---|---|---|---|---|---|
| InternVL3.5-4B | 82.2 | 86.4 | 77.8 | 65.2 | 88.9 | 2272 | 80.3 | 66.3 | 66.6 | 100.0% |
| + ViCO | 83.0 | 85.2 | 77.6 | 65.5 | 88.6 | 2239 | 80.5 | 66.5 | 65.9 | 99.7% |
| *Ratio of Compressed Patches* | *73%* | *62%* | *54%* | *64%* | *80%* | *76%* | *73%* | *76%* | *69%* | *69.7%* |
| InternVL3.5-8B | 84.0 | 86.5 | 77.6 | 68.5 | 88.7 | 2380 | 79.5 | 67.5 | 73.4 | 100.0% |
| + ViCO | 83.9 | 86.7 | 77.8 | 67.5 | 88.4 | 2381 | 79.8 | 66.8 | 71.9 | 99.6% |
| *Ratio of Compressed Patches* | *71%* | *47%* | *58%* | *67%* | *81%* | *77%* | *75%* | *75%* | *71%* | *69.1%* |
| InternVL3.5-14B | 83.2 | 86.1 | 77.3 | 67.7 | 87.7 | 2398 | 81.5 | 70.5 | 73.3 | 100.0% |
| + ViCO | 84.3 | 86.2 | 77.7 | 67.7 | 87.7 | 2392 | 81.0 | 71.0 | 73.2 | 100.2% |
| *Ratio of Compressed Patches* | *71%* | *44%* | *64%* | *66%* | *82%* | *76%* | *75%* | *79%* | *70%* | *69.7%* |
| InternVL3.5-30B-MoE | 88.1 | 87.5 | 80.2 | 71.6 | 89.6 | 2461 | 84.8 | 72.3 | 75.6 | 100.0% |
| + ViCO | 87.7 | 87.4 | 79.6 | 70.6 | 89.7 | 2462 | 83.8 | 71.8 | 75.9 | 99.6% |
| *Ratio of Compressed Patches* | *71%* | *62%* | *55%* | *69%* | *82%* | *76%* | *75%* | *64%* | *74%* | *69.8%* |
| InternVL3.5-38B | 88.0 | 88.9 | 82.8 | 71.6 | 90.4 | 2492 | 87.3 | 75.9 | 76.9 | 100.0% |
| + ViCO | 87.8 | 88.8 | 82.4 | 71.6 | 91.4 | 2496 | 86.8 | 74.8 | 75.0 | 99.6% |
| *Ratio of Compressed Patches* | *67%* | *32%* | *53%* | *56%* | *70%* | *70%* | *73%* | *73%* | *64%* | *62.0%* |
| InternVL3.5-241B-MoE | 91.1 | 88.6 | 84.6 | 74.3 | 90.7 | 2525 | 87.4 | 75.2 | 77.7 | 100.0% |
| + ViCO | 90.7 | 88.3 | 84.2 | 73.7 | 90.5 | 2527 | 87.2 | 74.6 | 76.9 | 99.6% |
| *Ratio of Compressed Patches* | *73%* | *55%* | *55%* | *60%* | *55%* | *62%* | *65%* | *77%* | *78%* | *64.4%* |

### 4.2.2 RESULTS ON OCR-RELATED BENCHMARKS

As shown in Table 1, the 8B model achieves a patch compression rate of 71% on OCRBench while retaining virtually the same performance (84.0 vs. 83.9). On ChartQA and TextVQA, the same model achieves 86.7 and 77.8, respectively, showing fluctuations compared to the original scores of 86.5 and 77.6. Across all OCR-related benchmarks, the performance remains largely consistent with the original models, demonstrating the robustness of our approach. This strong retention of performance can be attributed to ViCO's adaptive routing strategy. The model learns to compress simpler patches more aggressively, using lighter modeling, while preserving the original complex modeling for patches containing critical semantic information. As a result, ViCO enables the model to focus on semantically important regions without losing essential details, allowing high-fidelity performance even under substantial token compression.

### 4.2.3 RESULTS ON MULTI-IMAGE AND VIDEO UNDERSTANDING BENCHMARKS

As shown in Table 2, ViCO achieves substantial token compression while preserving performance. For example, on the Video-MME benchmark with subtitles, the largest 241B-MoE model compresses approximately 70% of the tokens but still achieves a score of 76.6, slightly surpassing the original model. Similar trends are observed across other models: the 38B model compresses 63% of Video-MME tokens and maintains almost 100% of its performance, while smaller models retain over 99% of their original scores after being compressed.

Across all multi-image and video benchmarks, the average performance remains consistently high, demonstrating that ViCO effectively accelerates token processing without sacrificing accuracy. This is achieved through our adaptive token compression strategy, which reduces computation on less informative patches while preserving complex modeling for semantically important regions. As a result, the model efficiently handles long visual sequences and maintains robust performance on tasks that require understanding across multiple images or video frames.

### 4.3 COMPARISON WITH EXISTING METHODS

As shown in Table 3, we evaluate our method against two recent token reduction approaches, FastV and SparseVLM, using a comparable average compression ratio on general benchmarks. Unlike our approach, these baselines rely on manually pre-defined hyperparameters and cannot adapt compression dynamically across tasks. Consequently, although their overall scores decrease only slightly, they exhibit substantial performance drops on vision-sensitive benchmarks such as OCR. In contrast, our method automatically assesses the semantic importance of tokens, allowing it to apply stronger

Table 2: **Performance of ViCO on InternVL3.5 models across multi-image and video benchmarks.** For Video-MME, both the subtitle and non-subtitle settings are evaluated using 64-frame inputs.

| Model | BLINK | MMT-Bench | MMIU | Video-MME | | Avg. |
|---|---|---|---|---|---|---|
| | | | | no sub | with sub | |
| InternVL3.5-4B | 59.5 | 64.3 | 49.2 | 65.4 | 68.6 | 100.0% |
| + ViCO | 58.3 | 64.2 | 49.1 | 65.1 | 68.1 | 99.2% |
| *Ratio of Compressed Patches* | *74%* | *62%* | *66%* | *47%* | *47%* | *59.2%* |
| InternVL3.5-8B | 59.5 | 66.7 | 49.4 | 66.0 | 68.6 | 100.0% |
| + ViCO | 58.9 | 67.1 | 50.7 | 66.1 | 68.7 | 100.4% |
| *Ratio of Compressed Patches* | *74%* | *61%* | *56%* | *40%* | *40%* | *54.2%* |
| InternVL3.5-14B | 57.6 | 68.0 | 51.3 | 67.9 | 68.6 | 100.0% |
| + ViCO | 59.2 | 67.9 | 50.5 | 67.9 | 68.7 | 100.3% |
| *Ratio of Compressed Patches* | *77%* | *63%* | *67%* | *55%* | *55%* | *63.4%* |
| InternVL3.5-30B-MoE | 60.0 | 66.6 | 55.1 | 68.7 | 71.8 | 100.0% |
| + ViCO | 60.2 | 66.9 | 54.5 | 69.0 | 71.4 | 99.9% |
| *Ratio of Compressed Patches* | *72%* | *60%* | *56%* | *42%* | *42%* | *54.4%* |
| InternVL3.5-38B | 60.9 | 71.8 | 58.9 | 70.9 | 74.2 | 100.0% |
| + ViCO | 62.2 | 71.5 | 59.6 | 71.3 | 74.0 | 100.6% |
| *Ratio of Compressed Patches* | *64%* | *54%* | *51%* | *63%* | *63%* | *59%* |
| InternVL3.5-241B-MoE | 61.4 | 72.7 | 61.3 | 72.9 | 76.5 | 100.0% |
| + ViCO | 64.8 | 72.0 | 61.4 | 73.7 | 76.6 | 101.1% |
| *Ratio of Compressed Patches* | *61%* | *58%* | *55%* | *70%* | *70%* | *62.8%* |

Table 3: **Ablation and comparative study of ViCO across general and strongly OCR-dependent benchmarks.** Experiments are conducted on InternVL3.5-8B. For DocVQA and InfoVQA, the validation sets are used for performance evaluation.

| Model | DocVQA | ChartQA | InfoVQA | TextVQA | OCRBench | AI2D | MMStar | BLINK | Avg. |
|---|---|---|---|---|---|---|---|---|---|
| Vanilla | 91.3 | 86.5 | 79.1 | 77.6 | 84.0 | 84.0 | 68.5 | 59.5 | 100.0% |
| Vanilla + VIR | 57.9 | 78.0 | 69.4 | 71.1 | 75.1 | 83.5 | 64.7 | 54.7 | 87.9% |
| Vanilla w/o dynamic res. | 56.2 | 75.0 | 38.6 | 64.0 | 68.7 | 83.0 | 64.1 | 56.0 | 80.2% |
| FastV | 87.5 | 84.0 | 71.6 | 76.9 | 81.0 | 83.0 | 64.2 | 56.8 | 95.9% |
| SparseVLM | 75.9 | 84.9 | 55.2 | 76.5 | 77.7 | 84.1 | 66.4 | 54.5 | 91.2% |
| ViCO (All Compress) | 85.8 | 84.5 | 68.9 | 74.8 | 80.6 | 83.2 | 66.3 | 55.4 | 95.1% |
| ViCO (Random Compress) | 88.8 | 85.9 | 73.3 | 76.8 | 83.0 | 83.3 | 66.6 | 57.4 | 97.6% |
| ViCO (Image-level ViR) | 89.2 | 86.5 | 79.1 | 76.7 | 82.9 | 84.2 | 67.4 | 56.8 | 98.8% |
| *Ratio of Compressed Patches* | *69%* | *36%* | *12%* | *65%* | *78%* | *32%* | *75%* | *88%* | *56.9%* |
| ViCO | 90.8 | 86.7 | 78.6 | 77.8 | 83.9 | 83.7 | 67.5 | 58.9 | 99.6% |
| *Ratio of Compressed Patches* | *60%* | *47%* | *21%* | *58%* | *71%* | *42%* | *67%* | *74%* | *55.0%* |

compression on vision-insensitive tasks like BLINK while retaining more tokens on vision-critical benchmarks such as InfoVQA, thereby avoiding significant degradation in performance.

## 4.4 ABLATION STUDY

**Settings.** We conduct ablation experiments on the InternVL3.5-8B model. To evaluate the consistency training stage, we construct the Vanilla with VIR variant by applying the router decisions from the fully trained ViCO model to the original model. For the router training stage, we test three variants: (1) All Compression: all patches are compressed indiscriminately; (2) Random Routing: patches are routed randomly at the same compression ratios as the ViCO model; (3) Image-Level Routing: the router operates at the image level instead of the patch level. Additionally, we perform an experiment on the original InternVL3.5-8B model without dynamic resolution.

**Effect of Consistency Training.** As shown in Table 3, directly applying the final-stage ViCO router to the InternVL3.5 model causes a substantial performance drop (e.g., OCRBench drops from 84.0 to 75.1), indicating that the InternVL3.5 model cannot handle interleaved visual tokens at different compression rates. Disabling dynamic resolution in InternVL3.5 also reduces visual tokens significantly but severely degrades performance on vision-sensitive benchmarks. These results highlight

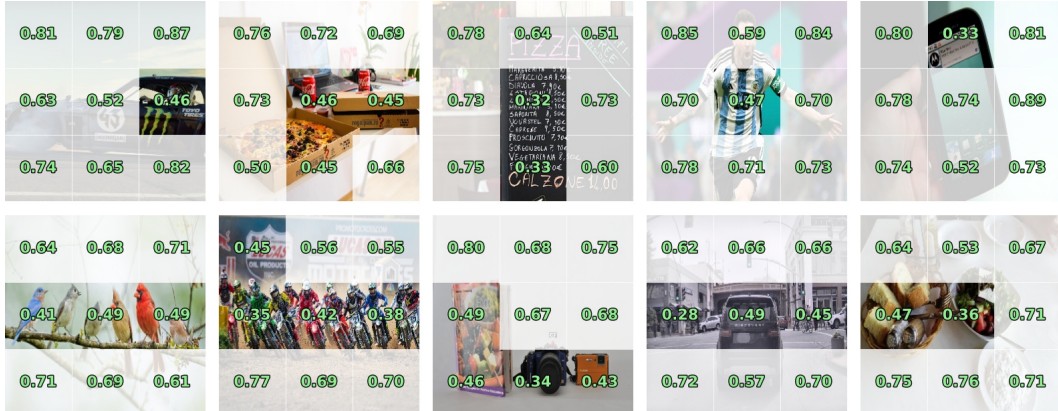

Figure 3: **Visualization of Routing Results of the InternVL3.5-8B ViCO Model.** White-shaded patches are compressed to 64 tokens, while unshaded patches retain 256 tokens. Each patch is annotated with the router's confidence score, where higher values indicate a stronger tendency for compression.

the necessity of Consistency Training, which allows the model to process interleaved tokens at varying downsampling rates while maintaining strong performance.

**Effect of Router Training.** As shown in Table 3, to validate the effectiveness of the router, we first evaluate a naive variant in which all visual tokens are compressed indiscriminately . This setting achieves a performance retention of 95.1%, indicating that uniform compression alone can harm model performance. Next, we test a variant where all patches are routed randomly to different compression rates while keeping the overall compression ratio identical to ViCO with VIR. This variant achieves 97.6% performance retention, still falling short of the 99.6% retained by our full ViCO with trained router. These results demonstrate that our router effectively learns to dynamically adjust patch-wise compression based on semantic content, with patches containing rich semantic information being lightly compressed, whereas patches with simpler content are more aggressively compressed. This learned adaptive routing enables a balanced trade-off between efficiency and model performance.

**Effect of Compression Granularity.** As shown in Table 3, we further investigate the impact of routing granularity by training a router that performs image-level compression instead of patch-level. In this variant, the router decides whether to compress all visual tokens of an image based on the overall image semantics. This image-level routing achieves 98.8% performance retention, slightly lower than the 99.6% obtained with patch-level routing. At the same time, image-level routing is overly coarse and lacks fine-grained semantic awareness. This causes the router to be highly sensitive to the dominant content of each image, leading to drastically different compression tendencies across datasets. For instance, only 12% of patches are compressed on InfoVQA, whereas 88% are compressed on BLINK. Such variability leads to unstable compression behavior. These results highlight the advantage of patch-level routing in ViCO, which achieves more balanced and semantically informed token compression.

## 4.5 VISUALIZATION OF IMAGE ROUTING

**Settings.** We visualize the routing results of our proposed Visual Resolution Router on several representative examples from the benchmarks introduced earlier. The routing is performed using the InternVL3.5-8B ViCO model, following the same evaluation settings as in the performance tests.Patches with router scores above 0.5 are treated as less critical and routed to higher compression (64 tokens), while those below 0.5 are routed to lower compression rates.

**Results.** As shown in Figure 3, our router is able to distinguish between semantically complex and simple patches. Regions containing salient objects, such as people, animals, or other key subjects, as well as areas with text carrying rich semantic information, are routed to lower compression rates and largely preserved at their original resolution. In contrast, relatively homogeneous background areas,

Table 4: **Throughput of LLM under different token compression rates.** Ori. refers to the original non-compressed model. We evaluate token compression rates of 25%, 50%, and 75% to measure the resulting speedup. All values are reported as speedup factors ($\times$) relative to the non-compressed baseline.

| Model | Input Type | LLM Throughput @ token compression | | | |
|---|---|---|---|---|---|
| | | Ori. | 25% | 50% | 75% |
| 8B | 10 patches + 64 text tokens | 1.00$\times$ | 1.29$\times$ | 1.97$\times$ | 3.76$\times$ |
| | 10 patches + 512 text tokens | 1.00$\times$ | 1.30$\times$ | 1.76$\times$ | 2.77$\times$ |
| 30B | 10 patches + 64 text tokens | 1.00$\times$ | 1.30$\times$ | 1.99$\times$ | 3.77$\times$ |
| | 10 patches + 512 text tokens | 1.00$\times$ | 1.29$\times$ | 1.80$\times$ | 2.72$\times$ |
| 38B | 10 patches + 64 text tokens | 1.00$\times$ | 1.33$\times$ | 1.99$\times$ | 3.80$\times$ |
| | 10 patches + 512 text tokens | 1.00$\times$ | 1.30$\times$ | 1.77$\times$ | 2.76$\times$ |
| 241B | 10 patches + 64 text tokens | 1.00$\times$ | 1.34$\times$ | 2.00$\times$ | 3.76$\times$ |
| | 10 patches + 512 text tokens | 1.00$\times$ | 1.32$\times$ | 1.81$\times$ | 2.87$\times$ |

which contain less critical information, are routed to higher compression rates. Importantly, being routed to higher compression does not imply that a patch is unimportant. Instead, it indicates that such information can be adequately represented with fewer resources. This selective compression strategy effectively reduces computational cost while allowing the model to focus more on patches that carry significant semantic content, ensuring that key visual information is retained for the LLM's processing. Additional visualizations are provided in Appendix C.

### 4.6 THROUGHPUT ANALYSIS

**Settings.** We deploy our ViCO models using the LMDeploy (Contributors, 2023) framework and evaluate different model scales, namely 8B, 30B, 38B, and 241B, under varying image–text ratios and compression rates. Using a real deployment framework ensures that the evaluation more accurately reflects practical inference performance. For throughput simulation, each request contains 10 visual patches with text inputs of length 512 or 64 tokens. The LLM's *first-token throughput* is measured over 2000 requests, which are sent concurrently to the API using 32 threads, and the results are reported as relative speedup compared to the non-compressed baseline. We set the tensor parallelism of the 241B model to 8, and to 1 for all other models.

**Results.** As shown in Table 4, applying token compression substantially improves the LLM's processing efficiency across all model scales and input settings. Specifically, compressing 50% of visual tokens consistently achieves over 1.75$\times$ speedup, while further increasing the compression ratio leads to even larger gains. Notably, smaller text input lengths benefit more from high compression than longer text inputs, as the relative proportion of compressed visual tokens is higher. These results demonstrate that our method not only reduces computational overhead theoretically but also yields significant practical acceleration during model inference, confirming the effectiveness of visual token compression for large-scale multimodal models.

## 5 CONCLUSION

In this work, we propose Visual Consistency Learning (ViCO), which enables the model to represent images of varying semantic complexities using different numbers of vision tokens. Compared with existing dynamic high-resolution strategies, which adjust the number of visual tokens based on image resolutions, our method dynamically adapts the number of visual tokens according to semantic complexity. Experimental results demonstrate that our method can reduce the number of vision tokens by up to 50% while maintaining the model's perception, reasoning, and OCR capabilities. Additional ablation studies further validate the effectiveness of each proposed module. By making decisions based on visual semantics, ViCO enables the model to efficiently focus computation on the most informative regions of an image, providing insights for future research on adaptive visual representations. We hope this work will contribute to the development of more efficient MLLMs. The code and models will be released to facilitate future research.

## ETHICS STATEMENT

This study complies with the ICLR ethical guidelines and adheres to the principles of responsible research. We confirm that no personally identifiable, sensitive, or potentially harmful data were utilized. We have considered the potential impact of our methods and believe that they advance scientific understanding without causing any foreseeable harm.

## REPRODUCIBILITY STATEMENT

We will open-source our code and model weights on platforms such as GitHub and Hugging Face, including methodological details and experimental settings, to ensure the reproducibility of our methods.

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

## A   THE USE OF LARGE LANGUAGE MODELS (LLMS)

Large language models (LLMs) were only employed for light linguistic refinement, such as polishing sentences and correcting grammatical errors. They were not involved in the formulation or development of the core ideas presented in this article.

## B   PERFORMANCE OF VICO ON SMALL-SCALE MODELS

We also evaluate ViCO on smaller InternVL3.5 models 1B and 2B across a range of benchmarks, including general multimodal, OCR-related, multi-image, and video benchmarks. Tables 5 and 6 show that the models retain nearly all of their original performance, demonstrating that our approach remains effective even on smaller-scale models.

Table 5: **Performance of ViCO on InternVL3.5 small-scale models across general benchmarks.** When calculating the overall score, MME is normalized from 0–2800 to 0–100. RWQA refers to RealWorldQA.

| Model | OCRBench | ChartQA | TextVQA | MMStar | POPE | MME | MMBench V1.1 | RWQA | Avg. |
|---|---|---|---|---|---|---|---|---|---|
| InternVL3.5-1B | 79.2 | 78.0 | 71.2 | 50.6 | 86.8 | 1910.2 | 69.9 | 57.6 | 100.0% |
| + ViCO | 78.8 | 77.4 | 71.1 | 50.8 | 86.2 | 1905.8 | 69.1 | 56.9 | 99.4% |
| *Ratio of Compressed Patches* | *70%* | *40%* | *56%* | *60%* | *80%* | *73%* | *72%* | *75%* | *65.8%* |
| InternVL3.5-2B | 83.7 | 80.8 | 76.7 | 57.5 | 87.2 | 2123.3 | 76.6 | 62.0 | 100.0% |
| + ViCO | 83.3 | 79.8 | 76.2 | 57.7 | 87.2 | 2101.0 | 76.7 | 60.7 | 99.4% |
| *Ratio of Compressed Patches* | *68%* | *36%* | *60%* | *61%* | *77%* | *73%* | *71%* | *66%* | *64.0%* |

Table 6: **Performance of ViCO on InternVL3.5 small-scale models across multi-image and video benchmarks.** For Video-MME, both the subtitle and non-subtitle settings are evaluated using 64-frame inputs.

| Model | BLINK | MMT-Bench | MMIU | Video-MME | | Avg. |
|---|---|---|---|---|---|---|
| | | | | no sub | with sub | |
| InternVL3.5-1B | 44.0 | 54.5 | 45.2 | 52.4 | 55.0 | 100.0% |
| + ViCO | 43.9 | 54.3 | 43.9 | 52.9 | 54.9 | 99.5% |
| *Ratio of Compressed Patches* | *69%* | *57%* | *56%* | *63%* | *63%* | *61.6%* |
| InternVL3.5-2B | 51.3 | 58.5 | 44.9 | 58.3 | 61.2 | 100.0% |
| + ViCO | 51.3 | 58.7 | 45.5 | 59.1 | 61.4 | 100.6% |
| *Ratio of Compressed Patches* | *69%* | *58%* | *48%* | *73%* | *73%* | *64.2%* |

## C   VISUALIZATION OF ROUTER

As shown in Figures 4 and 5, we visualize the Visual Resolution Router from the InternVL3.5-8B ViCO model, demonstrating additional routing results across different patch layouts.

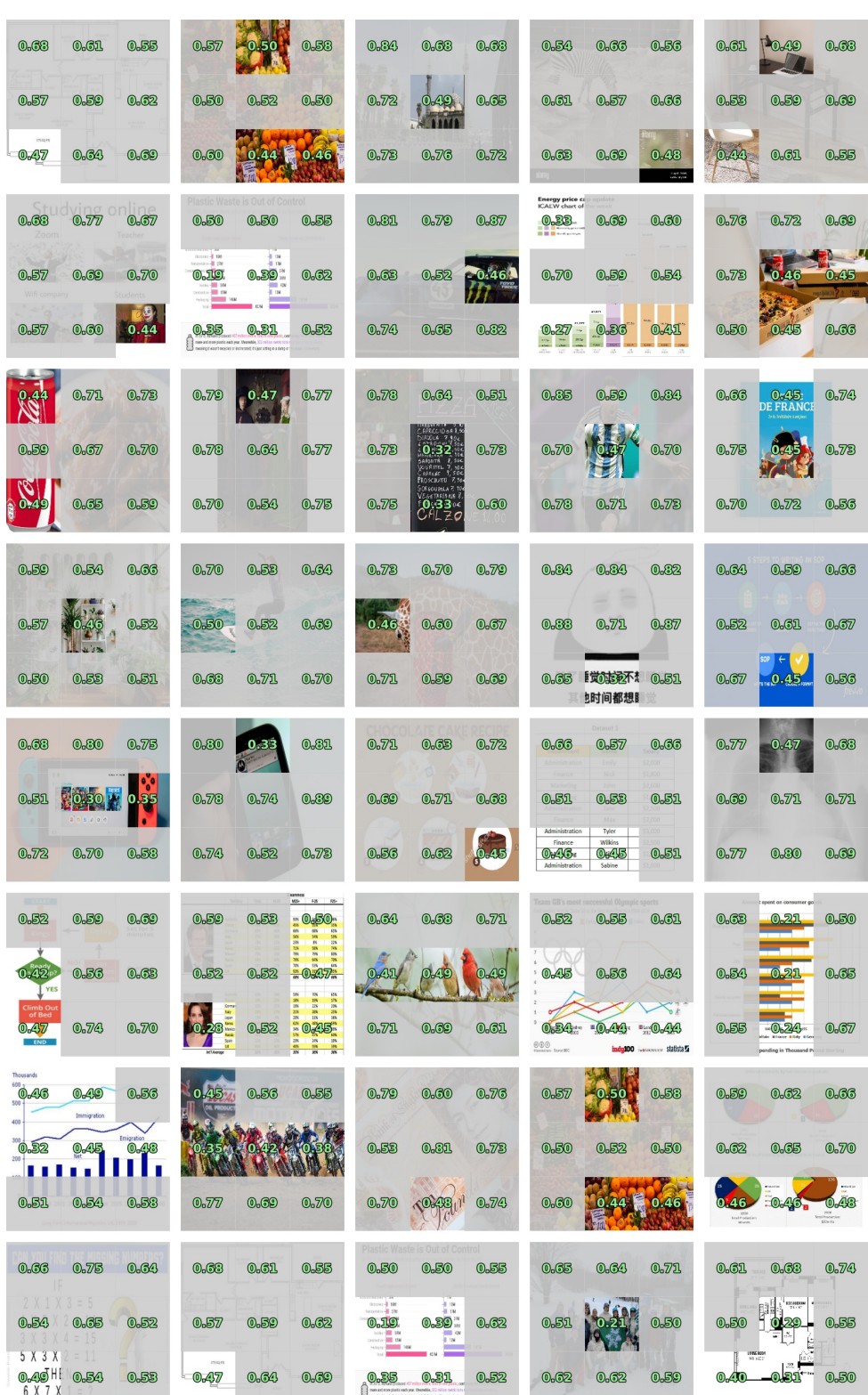

Figure 4: **Routing results of the InternVL3.5-8B ViCO model on images split into** $3 \times 3$ **patches.** Grey-shaded patches are compressed to 64 tokens, while unshaded patches retain 256 tokens. Each patch is annotated with the router's confidence score, where higher values indicate a stronger preference for compression.

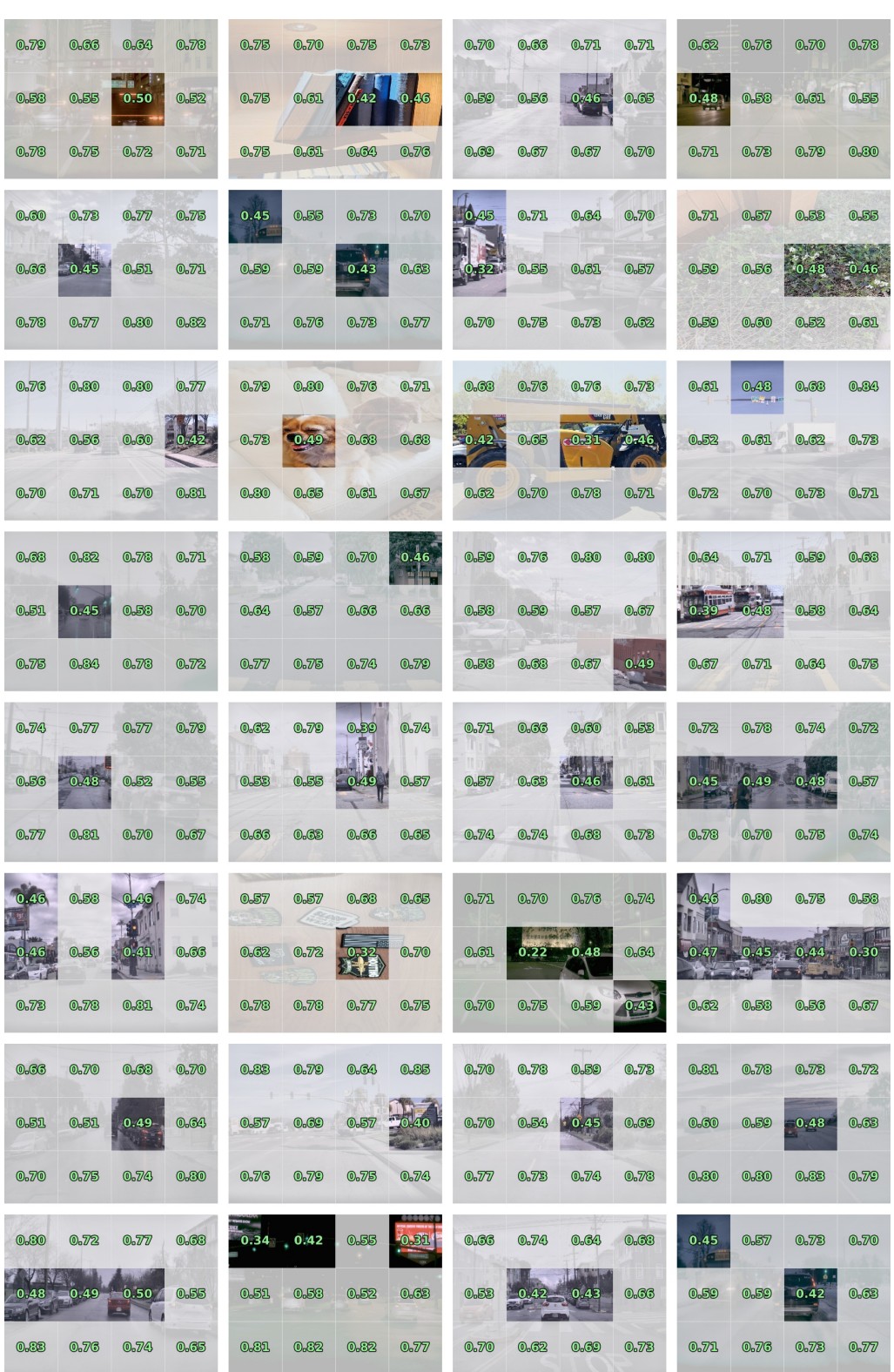

Figure 5: **Routing results of the InternVL3.5-8B ViCO model on images split into** $3 \times 4$ **patches.** Grey-shaded patches are compressed to 64 tokens, while unshaded patches retain 256 tokens. Each patch is annotated with the router's confidence score, where higher values indicate a stronger preference for compression.

