# OpenReview forum: "ViCO: A Training Strategy towards Semantic Aware Dynamic High-Resolution"
_ICLR.cc/2026/Conference — ICLR 2026 Conference Withdrawn Submission_

### Official Review · Reviewer_yrZR · 2025-10-26

**Soundness:** 1
**Presentation:** 3
**Contribution:** 3
**Rating:** 2
**Confidence:** 5

**Summary:**

To address the efficiency issues caused by the excessive number of visual tokens in existing MLLMs, the paper proposes the ViCo algorithm. The ViCo algorithm dynamically compresses different image patches into varying numbers of tokens. It comprises two primary stages: (1) a consistency training phase that enables the model to effectively adapt to varying compression ratios, and (2) the training of a Visual Resolution Router (ViR) that dynamically determines the optimal compression rate for each image input. Extensive experiments and visualizations demonstrate the effectiveness of ViCo.

**Strengths:**

1.	The paper introduces a consistency training method that enables the model to effectively adapt to different compression rates. Experimental results demonstrate that this method plays a crucial role in maintaining performance under compression.
2.	The paper introduces a ViR module, which enables the model to dynamically assign different compression rates to different image patches. Experimental results and visualizations demonstrate its effectiveness.
3.	The proposed method is validated across a wide range of benchmarks and MLLMs of different scales. Extensive ablation studies further demonstrate the effectiveness of the approach.

**Weaknesses:**

1.	A key limitation of ViCo lies in its reliance on initializing from an existing MLLM and requiring additional training. This implies that ViCo cannot accelerate the training of MLLMs and instead demand substantial extra training time, which contradicts the original intention of visual compression.
2.	The paper only compares ViCo with training-free compression methods (FastV and SparseVLM), without including more recent trainable compression approaches that can accelerate model training, such as TokenPacker [1] and Falcon [2]. It is worth noting that all these methods do not require additional training time.
3.	The paper evaluates ViCo only on the InternVL3.5 series, lacking validation on other mainstream open-source models such as Qwen2-VL [3] and LLaVA-OneVision [4]. Please note that it would be a critical issue if the paper intends to claim its contribution lies in “distilling” existing models into higher compression ratios rather than directly accelerating training.
4.	Previous studies have also explored similar dynamic compression methods, such as Avg-LLaVA [5] and Focus-LLaVA [6]. The paper does not include a comparative analysis.

[1] Li W, Yuan Y, Liu J, Tang D, Wang S, Qin J, Zhu J, Zhang L. Tokenpacker: Efficient visual projector for multimodal llm. International Journal of Computer Vision. 2025 Jun 27:1-9.

[2] Zhang R, Shao R, Chen G, Zhang M, Zhou K, Guan W, Nie L. Falcon: Resolving visual redundancy and fragmentation in high-resolution multimodal large language models via visual registers. In Proceedings of the IEEE/CVF International Conference on Computer Vision. 2025.

[3] Wang P, Bai S, Tan S, Wang S, Fan Z, Bai J, Chen K, Liu X, Wang J, Ge W, Fan Y. Qwen2-vl: Enhancing vision-language model's perception of the world at any resolution. arXiv preprint arXiv:2409.12191. 2024 Sep 18.

[4] Li B, Zhang Y, Guo D, Zhang R, Li F, Zhang H, Zhang K, Zhang P, Li Y, Liu Z, Li C. LLaVA-OneVision: Easy Visual Task Transfer. Transactions on Machine Learning Research.2025.

[5] Lan Z, Niu L, Meng F, Li W, Zhou J, Su J. AVG-LLaVA: An Efficient Large Multimodal Model with Adaptive Visual Granularity. In Findings of the Association for Computational Linguistics: ACL 2025 2025 Jul (pp. 16852-16869).

[6] Zhu Y, Xie C, Liang S, Zheng B, Guo S. Focusllava: A coarse-to-fine approach for efficient and effective visual token compression. arXiv preprint arXiv:2411.14228. 2024 Nov 21.

**Questions:**

1.	What are the data amount, data types, and training time needed for ViCo?
2.	Could authors explain why Consistency Training employs KL divergence between the reference model and the policy model for training, rather than directly supervising the policy model using the ground truth?

---

### Official Review · Reviewer_wnra · 2025-10-30

**Soundness:** 3
**Presentation:** 3
**Contribution:** 2
**Rating:** 4
**Confidence:** 4

**Summary:**

This paper proposes Visual Consistency Learning (ViCO), a training strategy for Multimodal Large Language Models (MLLMs) that dynamically adjusts the number of vision tokens based on the semantic complexity of image patches. ViCO employs multiple MLP connectors with different compression ratios and introduces a Visual Resolution Router (ViR) to select the appropriate token count per patch during inference. Through a two-stage training process—consistency training and router training, ViCO reduces visual tokens by up to 50% while preserving strong performance.

**Strengths:**

1. The two-stage process is reasonable and persuasive, and it possesses strong interpretability.
2. Experimental results show that ViCO achieves up to 50% visual token compression while maintaining performance (average >99.6%).

**Weaknesses:**

1. The verification of generalizability is inadequate, as all experiments are based on the InternVL3.5 model family and have not been validated on other MLLM architectures (such as LLaVA, BLIP, etc.). Additionally, this method requires the ViT to output fixed-length visual tokens, but current models like Qwen2.5-VL use native resolution ViTs, which produce variable-length visual tokens. ViCO does not seem applicable to these models.
2. There is a lack of comparison with other methods. The paper only compares FastV and SparseVLM, but there are many comparable methods in the field of visual token compression, such as [1][2][3].
[1] ToFu: Visual Tokens Reduction via Fusion for Multi-modal, Multi-patch, Multi-image Tasks.
[2] VoCo-LLaMA: Towards Vision Compression with Large Language Models.
[3] Stop Looking for “Important Tokens” in Multimodal Language Models: Duplication Matters More.
3. This paper introduces a routing mechanism and multiple MLPs. It would be better if the authors could analyze whether the additional structures would introduce extra computational overhead and additional inference latency.

**Questions:**

ViCO requires the additional overhead of two-stage training, but can its performance significantly outperform current training-free visual token compression methods?

---

### Official Review · Reviewer_Ccac · 2025-10-31

**Soundness:** 2
**Presentation:** 2
**Contribution:** 2
**Rating:** 4
**Confidence:** 4

**Summary:**

This paper proposes VICO (Visual Consistency Learning), a two-stage training strategy designed to reduce the high inference costs of Multimodal Large Language Models (MLLMs), which stem from the large number of visual tokens introduced by image inputs. Experimental results demonstrate that VICO can compress over 50% of visual tokens across various benchmarks (general, OCR, and video) while retaining over 99.6% of the original model's performance and significantly improving inference throughput.

**Strengths:**

The paper addresses a highly practical and important problem. As MLLMs process higher-resolution and more numerous images (e.g., in documents and videos), the overhead from visual tokens has become a primary inference bottleneck.

 The two-stage design of VICO is logical. First, using "Consistency Training" to make the model adapt to information loss, and then training a router to make decisions, is demonstrably more effective than applying a router directly

**Weaknesses:**

The paper's core motivation is to route based on "semantic complexity." However, the method's actual implementation (Eq 2) does not measure objective semantic complexity. Instead, it measures the model's own sensitivity to compression (i.e., the change in the $\mathcal{L}_{ViCO}$ loss).

VICO is a two-stage training method that requires significant computational resources (including fine-tuning the entire MLLM for consistency and then training the router).

To fairly evaluate VICO's contribution, the authors should compare it against other dynamic token allocation or pruning methods that also require training. The current comparison (VICO vs. training-free methods) demonstrates that "training helps" more than it demonstrates that "VICO's method is superior" to other trained approaches.

Dynamic token routing and pruning are not new concepts. While the "Consistency Training" (Stage 1) is a valuable contribution, the "Router Training" (Stage 2) is, methodologically, a standard classifier: extract patch features (ViT + AttnPool) and perform binary classification (compress/don't compress). Its core novel component—the pseudo-label generation (Eq 2)—is, as noted in W1, methodologically flawed.

**Questions:**

see the weakness

---

### Note · Authors · 2025-11-12

I have read and agree with the venue's withdrawal policy on behalf of myself and my co-authors.